# Presence of N, N′-Substituted p-Phenylenediamine-Derived Quinones in Human Urine

**DOI:** 10.3390/toxics12100733

**Published:** 2024-10-11

**Authors:** Juxiu Huang, Hangbiao Jin, Yingying Zhu, Ruyue Guo, Lisha Zhou, Xiaoyu Wu

**Affiliations:** 1Taizhou Central Hospital (Taizhou University Hospital), School of Medicine, Taizhou University, Taizhou 318000, China; huangjx7163@tzzxyy.com (J.H.); lishazhou@tzc.edu.cn (L.Z.); 2Key Laboratory of Microbial Technology for Industrial Pollution Control of Zhejiang Province, College of Environment, Zhejiang University of Technology, Hangzhou 310032, China; hangbiao102@163.com (H.J.); gryue1999@163.com (R.G.); 3School of Life Sciences, Taizhou University, Taizhou 318000, China; zhuyingying1988@tzc.edu.cn

**Keywords:** p-phenylenediamine-derived quinones, 6PPDQ, CPPDQ, human urine, human exposure

## Abstract

Human exposure to various N,N′-substituted p-phenylenediamine-derived quinones (PPDQs) has been of increasing concern. Recent studies have examined N-phenyl-N′-(1,3-dimethylbutyl)-p-phenylenediamine-derived quinone (6PPDQ) in human urine to evaluate human exposure. However, other PPDQs in human urine have not been thoroughly investigated. This study analyzed six PPDQs in urine collected from 149 healthy individuals in Taizhou, China. All target PPDQs were detected, with 6PPDQ (mean 2.4 ng/mL, <limit of detection (LOD)–19 ng/mL) and 2-(cyclohexylamino)-5-(phenylamino)cyclohexa-2,5-diene-1,4-dione (CPPDQ; 2.1 ng/mL, <LOD–24 ng/mL) being the most prevalent. Human urinary concentrations of 2,5-bis((5-methylhexan-2-yl)amino)cyclohexa-2,5-diene-1,4-dione (77PDQ; mean 1.5 vs. 0.87 ng/mL; *p* = 0.013) and 2,5-bis(o-tolylamino)cyclohexa-2,5-diene-1,4-dione (mean 1.1 vs. 0.62 ng/mL; *p* = 0.027) were significantly higher in females compared to males. For CPPDQ (*p* < 0.01) and 6PPDQ (*p* < 0.01), a decrease was observed in urinary concentrations as participants aged. The daily excretion (DE) of PPDQs through urine was estimated for Chinese adults. The highest average DE was recorded for 6PPDQ at 81 ng/kg-bw/day, with a range from <0.5 to 475 ng/kg-bw/day. Following this, CPPDQ had a mean DE of 68 ng/kg-bw/day (range <0.5–516 ng/kg-bw/day), and 77PDQ had a mean DE of 30 ng/kg-bw/day (<0.5–481 ng/kg-bw/day). This study is the first to explore the presence of various PPDQs in human urinary samples, which is essential for assessing the potential health risks associated with these substances.

## 1. Introduction

N,N′-substituted p-phenylenediamines (PPDs) represent a prominent group of synthetic compounds incorporated into various rubber-related products for over thirty years to avert rapid aging, oxidative degradation, and cracking [1,2,3]. N-isopropyl-N′-phenyl-1,4-phenylenediamine (IPPD), N-phenyl-N′-cyclohexyl-p-phenylenediamine (CPPD), and N-phenyl-N′-(1,3-dimethylbutyl)-p-phenylenediamine (6PPD) are the most commonly encountered kinds of PPDs that are employed in automotive wheels, or rubber tubes, as well as different types of rubber goods [4,5,6]. By 2020, China’s capacity for 6PPD production exceeded 200 kilotons annually, and the pervasive application of these PPDs has resulted in their broad environmental dissemination [7,8,9,10]. Additionally, the oxidation of 6PPD might lead to the formation of a quinone by-product called N-phenyl-N′-(1,3-dimethylbutyl)-p-phenylenediamine-derived quinone (6PPDQ) in the natural environment, as reported by Tian et al. [11]. This substance exhibits significant toxicity to coho salmon. Given the structural similarity among different PPDs, quinone derivatives from other PPDs might pose similar toxicological risks to biota, raising global scientific concerns regarding the environmental prevalence and toxicological impact of PPD-derived quinones (PPDQs) [3,12,13,14].

Environmental monitoring has confirmed the presence of several PPDQs in various matrices in the environment, including soil, dust, sediment, runoff water, and surface water [13,15,16,17]. Predominantly detected PPDQs include 6PPDQ, IPPD-quinone (IPPDQ), and CPPD-quinone (CPPDQ). Given their ubiquitous presence, the general population is unavoidably exposed to these compounds via the inhalation of dust, dermal contact, and dietary intake [3,18]. Environmental concentrations of PPDQs are often comparable to or exceed those of their parent PPD compounds. For example, concentrations of 6PPDQ (median 1.7–6.7 pg/m^3^) in PM_2.5_ samples from six Chinese cities are similar to those of 6PPD (0.9–8.4 pg/m^3^) [19]. Furthermore, Lyu et al. [20] suggested that 6PPDQ exposure may induce oxidative stress and DNA damage in lung tissue. Recent findings indicate that 6PPDQ could disrupt lipid metabolism and trigger inflammatory responses in the liver of mice [21], implying that PPDQ exposure may adversely affect human health. These findings highlight the necessity of studying PPDQ exposure among the general population.

Previous studies have attempted to estimate human PPDQ intake. For example, [16] estimated daily PPDQ intake through PM_2.5_ inhalation for juveniles, local residents, and adults, ranging between 0.16 and 1.25 ng/kg-bw/day, suggesting that the actual total human PPDQ exposure might still have been underestimated. Similarly, Cao et al. [15] discovered that individuals in Hong Kong, China, consumed a daily range of 1.1 to 7.3 ng/kg-bw/day of PPDQs. Despite these estimates, the precise quantification of human PPDQ exposure remains elusive.

Urine analysis offers an alternative approach to estimating PPDQ intake, as urine is a primary excretion route for various environmental pollutants. Human urine has been used as a biomarker in the assessment of human exposure to many pollutants, including heavy metals, polycyclic aromatic hydrocarbons, and bisphenol analogs [22,23,24]. To date, research conducted by Du et al. [25] was the first to investigate the concentrations of 6PPD and 6PPDQ in samples of urine collected from the Chinese population, reporting higher excretion of 6PPDQ (median 2.18–90.9 ng/kg-bw/day) compared to 6PPD (0.51–2.13 ng/kg-bw/day). Until now, the occurrence of PPDQs (except 6PPDQ) in urine from different human populations has not been well investigated and further research is necessary to clarify this.

In this study, we recruited 149 individuals from the general population living in Taizhou, China, and collected their urine samples. The occurrence and concentration profiles of six kinds of PPDQs were characterized in the collected human urine samples. Urinary concentrations of PPDQs in different age or gender groups were compared. In addition, the total amount of PPDQs excreted in human urine was estimated. These results contribute to a deeper understanding of human exposure to PPDQs, essential for assessing the associated health risks.

## 2. Materials and Methods

### 2.1. Standards and Reagents

Analytical standards of PPDQs used for analysis, such as 6PPDQ (2-((4-methylpentan-2-yl)amino)-5-(phenylamino)cyclohexa-2,5-diene-1,4-dione), DTPDQ (2,5-bis(*o*-tolylamino)cyclohexa-2,5-diene-1,4-dione), CPPDQ (2-(cyclohexylamino)-5-(phenylamino)cyclohexa-2,5-diene-1,4-dione), IPPDQ (2-(isopropylamino)-5-(phenylamino)cyclohexa-2,5-diene-1,4-dione), 77PDQ (2,5-bis((5-methylhexan-2-yl)amino)cyclohexa-2,5-diene-1,4-dione), and DPPDQ (2,5-bis(phenylamino)cyclohexa-2,5-diene-1,4-dione), were procured from suppliers such as Anhuida Chemical (Zhengzhou, China), Viti Chemical (Xinxiang, China), and J&K Scientific (Beijing, China). The isotopically labeled standard (6PPDQ-*d*_5_), serving as an internal standard, was purchased from Cambridge Isotope Laboratories (Andover, MA, USA). The full names along with abbreviations of the PPDQs studied are detailed in Appendix A. Solvents and reagents, including methanol, acetonitrile, ultrapure water, ammonium hydroxide, and formic acid, were from Sigma-Aldrich (Ontario, Canada).

### 2.2. Study Population and Sample Collection

Participants were recruited from the urban population of the city of Taizhou during the period from July to August 2024. All participants were local residents who had lived in urban areas of Taizhou for over twenty-four months, without professional exposure to the target PPDQs. The city of Taizhou, located in the eastern region of China, has approximately 6.2 million inhabitants [14]. It is a rapidly developing urban area with a mix of industrial, agricultural, and residential zones. The city has a growing industrial base, particularly in the manufacturing sector, which includes electronics, textiles, and machinery. However, as noted, to our knowledge, there are no specific chemical factories producing PPDs and PPDQs in the region. This study recruited a total of 149 healthy adult participants. The inclusion criteria for our study were: (1) participants had to be healthy adults aged 18 years or older, (2) they had to be local residents of Taizhou, having lived in the urban area for at least 24 months, and (3) participants had to have provided informed consent. The exclusion criteria included: (1) individuals with occupational exposure to the target chemicals (PPDQs) and (2) individuals with any known chronic diseases or conditions that could potentially affect the study’s outcomes. These criteria were established to ensure a focus on the general population and avoid confounding factors related to professional exposure or health conditions. Among these individuals, around 45% were males. The body mass index of the participants averaged 28 ± 6.1 kg/m^2^. The mean (± SD) age of female participants was 47 ± 14 years and for male participants, it was 46 ± 11 years. Approximately 55% of the participants reported a household income ranging from 80,000 to 150,000 CNY. The detailed demographic characteristics of the recruited participants were obtained through a questionnaire survey, as presented in Appendix A.

Each participant contributed a single urine sample, collected in the morning after fasting (approximately 12 mL). Trained nurses at Taizhou Central Hospital were responsible for the collection process. The collected human urine samples were kept frozen at −80 °C until they were ready for extraction. In addition, control samples (field blanks) consisting of 12 mL of distilled water were prepared at the collection site concurrently with the human urine samples. This research was conducted with the approval of the ethics committee at Taizhou Central Hospital, and written informed consent was obtained from all participants before they were enrolled in the study.

### 2.3. Sample Extraction

The extraction procedure for PPDQs from human urine samples was adapted from Cao et al. [15], with some adjustments. In summary, 2.0 mL of human urine was spiked with the internal standard. The mixture was then passed through HLB cartridges (250 mg/6 mL; Oasis, Waters Co., Milford, MA, USA) by gravity. The solid-phase extraction cartridges were initially conditioned using 6 mL of methanol, followed by 6 mL of Milli-Q water. Once these samples were loaded, the PPDQs trapped by the HLB cartridges were flushed out with a 6 mL solution of methanol containing 0.1% NH_4_OH. The collected eluents were then subjected to evaporation to achieve dryness under a stream of nitrogen gas. The remaining residue was then re-dissolved in 50 μL of a mixture consisting of 50% methanol and 50% water.

### 2.4. Instrumental Analysis

For the analysis of target PPDQs, a Waters ACQUITY I Class system was employed in conjunction with a XEVO TQ-S triple quadrupole mass spectrometer, sourced from Waters Co. (Milford, MA, USA). Chromatographic isolation was achieved using a Waters ACQUITY HSS T3 column (1.7 μm, 2.1 mm × 100 mm) at a constant temperature of 40 °C. A gradient elution method was utilized, starting with a mix of 20% methanol (phase B) and 80% pure water with 0.1% formic acid (phase A) for the initial 0.5 min. The proportion of methanol was then increased to 40% by the first minute, and further ramped up to 100% by the 10 min mark. This condition was maintained for 4 min before swiftly reverting to the starting mix of 20% methanol. All mass spectrometry data were collected in the positive ion mode using an electrospray ionization source. The instrument was operated under the multiple reaction monitoring (MRM) mode to obtain the mass spectral data. Specific MRM transitions for each PPDQ can be found in Appendix A.

### 2.5. Daily Excretion (DE) Calculation

Based on the measured levels of PPDQs in human urine and following the methodologies from previous research [25,26,27], the DE of PPDQs in human urine, expressed in ng/kg-bw/day, was calculated. The following equation was used to calculate the DE value.
$$DE=\frac{C_{PPDQs}\times V_{urine}}{BW}$$

In this equation, *C*_PPDQs_ represents the levels of specific PPDQ found in the collected urine samples, measured in ng/mL. The term *V*_urine_ refers to the daily volume of urine excretion for Chinese adults, which was estimated to be 1700 mL/day according to previous research [25,28]. B*W* represents the body mass of Chinese adults (kg-bw), and these data were obtained from the participants through a detailed questionnaire survey. It is important to note that this DE estimate has limitations. One major constraint is the reliance on the concentrations of PPDQs found in individual morning urine samples. This is because there can be daily variations in the levels of PPDQs in human urine. Despite these potential fluctuations, prior research supports the use of early morning urine samples as a feasible method for estimating human exposure to pollutants, and this is mainly due to their ability to consistently reflect the body’s exposure levels over time [29,30].

### 2.6. QA/QC

Background contamination by PPDQs was thoroughly checked in all utilized solvents. To ensure the accuracy of the analysis and to monitor potential carry-over and background contamination, each set of ten samples was accompanied by one solvent blank (10 μL of high-purity methanol), one procedural control blank, and one sample blank. No obvious target PPDQs were detected in these blank samples. The sampling kits for human urine sample collection were rinsed with pure methanol to minimize background contamination by PPDQs.

We measured the levels of PPDQs present in the human urine specimens by employing the internal standard approach. Calibration curves for individual PPDQs included six concentration points (0.5–200 ng/mL), and all of the correlation coefficients (R^2^) for the calibration curves were >0.995. We defined the limits of detection (LODs) for the target PPDQs. These LODs were established based on human urinary concentrations that produced a signal-to-noise ratio of 3.0 [16]. LOQs (limits of quantification) were established based on human urinary concentrations that produced a signal-to-noise ratio of 10. Appendix A shows that the calculated LODs for the target PPDQs ranged from 0.017 ng/mL for DPPDQ to 0.097 ng/mL for DTPDQ. To evaluate the extraction recovery of the PPDQs, we analyzed fortified human urine samples (*n* = 5).We spiked these samples with the target PPDQs at three different concentrations: 0.5 ng/mL, 5.0 ng/mL, and 50 ng/mL. Recovery for these PPDQs ranged from 82% for IPPDQ to 107% for DPPDQ. Details on the extraction recovery are provided in Appendix A. The reported PPDQ levels in the collected urine samples did not account for recovery adjustments. Repeatability was assessed by calculating the relative standard deviation (RSD). This evaluation involved determining the target PPDQs in fortified human urine samples with concentrations of 5.0 ng/mL or 20 ng/mL (*n* = 5). Measurements were conducted either on the same day (intra-day) or over two weeks (inter-day). The RSD values of intra-day and inter-day measurements for PPDQs were 7.7–14% and 9.2–17%, respectively. The impact of the matrix on the analyte signal was evaluated by adding native analyte standards to a blank human urine matrix at concentrations of 1.0 ng/mL and 15 ng/mL. The responses were then compared to those of native standards prepared in methanol. The matrix effect for the target analytes was determined to range between 97% and 105%.

### 2.7. Statistical Analysis

To investigate the relationships among urinary concentrations of various PPDQs and their correlation with participants’ ages, Spearman’s rank correlation was applied. Differences in urinary PPDQ levels between genders, as well as variations in the DE of different PPDQs, were assessed using the Mann–Whitney *U* test. All data analyses were performed with the SPSS software (version 26, IBM; Cambridge, MA, USA). Statistical significance was determined by a *p* value of less than 0.05 (two-tailed).

## 3. Results

### 3.1. Concentrations of PPDQs in Human Urine

Table 1 shows that all target PPDQs were found in human urine from Taizhou, China, with a detection frequency of 55% to 92%. The sum urinary concentrations of all detected PPDQs (∑PPDQs) were in the range of 1.7–34 ng/mL (mean 6.2 ng/mL). These findings align with the reported wide presence of PPDQs in dust, PM_2.5_, air particles, and runoff water samples [9,16,31,32]. The mean concentrations of 6PPDQ and CPPDQ in human urine were 2.4 ng/mL (<LOD–19 ng/mL) and 2.1 ng/mL (<LOD–24 ng/mL), respectively. These two PPDQs were the predominant PPDQs in human urine. They collectively constituted an average of 68% of the total detected PPDQs in human urine (Figure 1).

In this study, the median human urinary concentration of 6PPDQ was measured at 2.0 ng/mL. This value exceeds the median urinary 6PPDQ concentrations observed in adults and children living in Guangzhou, China, which were 0.40 ng/mL and 0.076 ng/mL, respectively [25]. However, it is lower than the median concentration of 2.91 ng/mL found in pregnant women [25]. Our study is the first to report concentrations of various PPDQs (except 6PPDQ) in human urine. This makes it hard to compare with previous data.

Among the PPDQs, significant correlations in urinary concentrations were only observed between 6PPDQ and CPPDQ (Spearman’s correlation coefficient, *r*_s_ = 0.73, *p* < 0.01), as well as between DTPDQ and DPPDQ (*r*_s_ = 0.59, *p* < 0.01) (Appendix A). This is inconsistent with the significant correlations always observed among different PPDQs in environmental matrix samples from China, such as sediment, dust, and PM_2.5_, [9,16]. This result indicates that humans may be exposed to these detected PPDQs through different sources, considering the wide presence of PPDQs in different environmental samples. This discrepancy is also possibly due to the different metabolic behaviors of these PPDQs in the human body. This is possible since the estimated log *K*_ow_ values of these detected PPDQs are relatively large, varying from 2.6 to 4.6 [15]. Alternatively, levels of PPDQs in human urine are also greatly influenced by the co-exposure and metabolism of their parent PPDs. Biological studies have demonstrated the biotransformation of 6PPD to 6PPDQ in zebrafish embryos (Danio rerio) and mice [21,33]. Considering their similarity in terms of chemical structure, other PPDs may also have similar, but still different, metabolic behaviors as 6PPD. Therefore, to find out the sources of PPDQs in the human body, further research is still needed.

### 3.2. Gender- and Age-Specific Differences

Across all participants, the mean urinary concentrations of each PPDQ were consistently higher in females than in males (Figure 2). However, significant differences were observed only for 77PDQ (mean 1.5 vs. 0.87 ng/mL; *p* = 0.013) and DTPDQ (1.1 vs. 0.62 ng/mL; *p* = 0.027). Du et al. [25] also reported significantly (*p* < 0.05) higher urinary 6PPDQ concentrations in women than men from Guangzhou, China. This gender-related difference may be due to factors such as the higher dermal exposure to PPDs from cosmetic products, rubber gloves, and underwear for females [34,35,36,37]. Additionally, the faster excretion rates of PPDQs in females compared to males may contribute to this difference [38].

Grouping individuals into four age categories (24–30, 31–40, 41–50, and 51–62 years groups) based on their age allowed us to examine the potential age-specific variations in urinary PPDQ concentrations (Figure 3). We observed a decreasing trend in urinary concentrations of CPPDQ (*r*_s_ = 0.69, *p* < 0.01) and 6PPDQ (*r*_s_ = 0.74, *p* < 0.01) with increasing age. The mean 6PPDQ concentration declined with age from 1.2 ng/mL (in the 24–30 years group) to 0.70 ng/mL (in the 51–62 years group). One possible explanation for this trend is the difference in the pharmacokinetics of PPDQs between younger and older adults. Specifically, the rate of renal clearance and biological metabolism of PPDQs may decrease with the increase in age [39,40,41], which may contribute to the higher urinary concentrations of PPDQs in younger adults. Alternatively, young adults may be exposed to more PPDs and PPDQs than old adults.

### 3.3. Human Daily Excretion (DE) of PPDQs

The DE of PPDQs through urine for Chinese adults is presented in Table 2. The DE of ∑PPDQs ranged from 2.1 to 1163 ng/kg-bw/day, averaging 198 ng/kg-bw/day. Of all the PPDQs, 6PPDQ exhibited the highest average daily excretion (81 ng/kg-bw/day, <0.5–475 ng/kg-bw/day), followed by CPPDQ (68 ng/kg-bw/day, <0.5–516 ng/kg-bw/day) and 77PDQ (30 ng/kg-bw/day, <0.5–481 ng/kg-bw/day). The lowest average estimated DE was observed for IPPDQ, with a mean of 4.4 ng/kg-bw/day, and a range spanning from <0.5 to 20 ng/kg-bw/day. Additionally, females showed a significantly greater (*p* < 0.05) mean daily excretion of various PPDQs (excluding IPPDQ) in comparison to males.

Du et al. [25] had estimated the median DE of 6PPDQ in urine for three different demographic groups in China: adults (11.3 ng/kg-bw/day), children (2.2 ng/kg-bw/day), and pregnant women (90.9 ng/kg-bw/day). The DE level of 6PPDQ for pregnant women reported in their study exceeds the values observed in our current research. The authors proposed that pregnant women may have a higher renal clearance of 6PPDQ or have specific exposure sources to 6PPDQ, such as more water consumption during pregnancy [10,42]. Wang et al. [16] estimated the amount of PPDQs inhaled via PM_2.5_ by children, adults, and workers to be between 0.16 and 1.25 ng/kg-bw/day. In another study, ref. [15] determined that people living in Hong Kong, China, orally ingested PPDQs in amounts ranging from 1.08 to 7.3 ng/kg-bw/day. These intake estimates are significantly lower than the DE of PPDQs found in our study, indicating that previous studies may have underestimated the overall amount of human exposure to PPDQs. Notably, the total amount of human exposure to PPDQs may still be underestimated in this study, since other ways of excretion for PPDQs (e.g., feces and exhaled air) and metabolism of PPDQs in the human body were not incorporated. Further research is required to investigate all potential pathways through which humans are exposed to various PPDQs.

### 3.4. Strengths and Limitations of This Study

One of the key strengths of our study is its comprehensive data collection, which provides valuable insights into the environmental implications of the target PPDQs we investigated. The meticulous methodology and focus on a specific population allow for a detailed understanding of the potential health impacts associated with exposure. This targeted approach not only contributes to the existing body of literature but also paves the way for future research by highlighting critical areas for further investigation. However, our study also has notable limitations. The absence of a control group restricts our ability to make definitive conclusions about causality, making it challenging to discern whether the observed effects are directly linked to the exposures of target PPDQs. Additionally, our sample size may limit the generalizability of the findings to broader populations, as it may not adequately represent diverse demographic groups. We recognize these limitations and are committed to addressing them in future research to enhance the validity and applicability of our conclusions.

## 4. Conclusions

This study examined the concentrations of various PPDQs in urine samples from adults living in Taizhou, China, marking the first occurrence of five kinds of PPDQs in human urine samples for the first time. The results indicate that humans had exposure to these compounds, with 6PPDQ and CPPDQ being the most predominant in the collected human urine. Further research is necessary to understand the risks associated with exposure to these PPDQs for the general population. Urinary levels of 77PDQ and DTPDQ were notably greater in female participants than in male participants, indicating a significant difference between the two gender groups. Additionally, a decreasing trend in the urinary concentrations of CPPDQ and 6PPDQ with increasing age was observed among the participants. The influence of age and gender on the metabolism of PPDQs in the human body needs further studies. Among the PPDQs, 6PPDQ had the highest mean DE, followed by CPPDQ and 77PDQ. Despite this estimation being preliminary, this DE can still reflect the internal human exposure dose of PPDQs, which is important for evaluating human exposure risks.

## Figures and Tables

**Figure 1 toxics-12-00733-f001:**
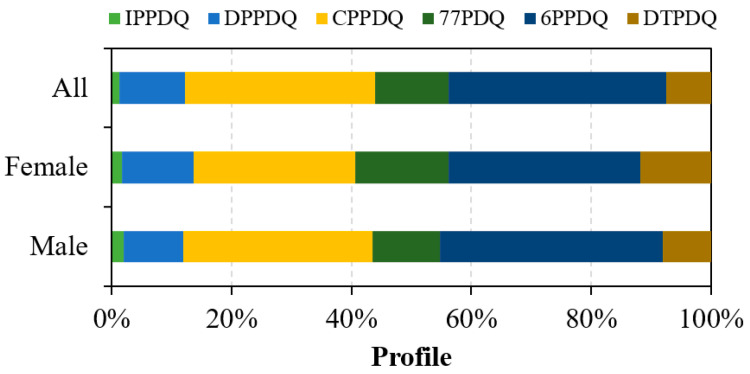
Composition profiles of PPDQs in human urine (*n* = 149) from Taizhou, China.

**Figure 2 toxics-12-00733-f002:**
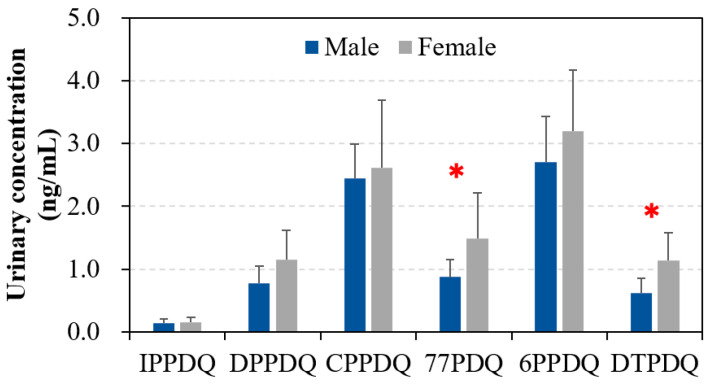
Concentrations (mean ± SD) of PPDQs in urine samples from male and female participants. The red asterisk indicates the significant (*p* < 0.05) difference in the human urinary concentration between male and female participants.

**Figure 3 toxics-12-00733-f003:**
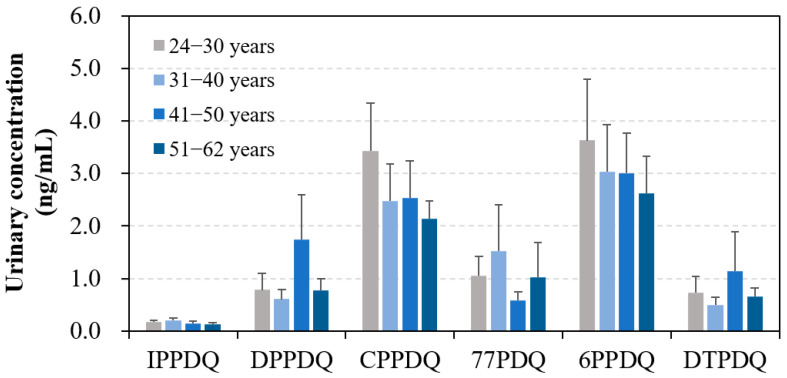
Concentrations (mean ± SD) of PPDQs in human urine from participants among different age groups.

**Table 1 toxics-12-00733-t001:** Urinary Concentrations (ng/mL) of PPDQs in Participants from Taizhou, China (*n* = 149).

	Detection Frequency	Mean	Median	Concentration Percentile			
				Min	25th	75th	Max
6PPDQ	92%	2.4	2.0	<LOD	0.83	3.4	19
CPPDQ	84%	2.1	1.6	<LOD	0.82	2.8	24
77PDQ	81%	0.81	0.42	<LOD	0.15	1.1	14
DPPDQ	79%	0.72	0.52	<LOD	0.20	0.98	11
DTPDQ	63%	0.49	0.37	<LOD	<LOD	0.92	8.9
IPPDQ	55%	0.082	0.11	<LOD	<LOD	0.28	0.65

**Table 2 toxics-12-00733-t002:** Estimated Daily Excretion (DE, ng/kg-bw/day) of PPDQs in Human Urine.

	Mean	Median	Range
IPPDQ	4.4	3.4	<0.5–20
DPPDQ	26	13	<0.5–338
CPPDQ	68	49	<0.5–516
77PDQ	30	11	<0.5–481
6PPDQ	81	54	<0.5–475
DTPDQ	22	10	<0.5–281

## Data Availability

Dataset can be made available on request to the authors.

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
