# Peer review of "Presence of N, N′-Substituted p-Phenylenediamine-Derived Quinones in Human Urine"

_toxics, 2024, doi:10.3390/toxics12100733_

Round 1
Reviewer 1 Report
Comments and Suggestions for Authors
Dear authors, after reading your paper, I have the following comments/questions:
1. How did you include the patients? What were the inclusion/exclusion criteria?
2. Any chance that they could be professionally exposed to the same chemicals?
3. Results: Are they good, are they bad? Hard to say since there is no control group
4. What are, in your opinion, the strengths and limitations of your study?
5. You can not say that your results indicate a widespread human exposure, since you have searched in only one city.
6. Can you give more data regarding Taizhou? E.g. are there any chemical factories that could influence your results?
Reviewer 2 Report
Comments and Suggestions for Authors
The paper manuscript “Presence of N, N’-Substituted p‑Phenylenediamine-Derived Quinones in Human Urine.”
Overall, this is a well written manuscript and has a potential to be accepted.
The study must summarize and clearly present the findings (regarding the degradation products) related to each method used. Furthermore, the study must examine how the findings relate to previous research in this area.
In addition, several comments follow.
1) General Comment: Please check abbreviations with consistency in main text. Define it at the first appearance, then use it after the definition.
2) Lines 123-132 Please justify the use of glutathione solution to urine sample extraction.
3) Lines 166-171: A paragraph of matrix effect assessment can be added.
3) Lines 177-178: Please provide a reference based on the calculation of LOD based on S/N. LOQ, calculation/estimation can be also added.
4) A scientifically justification and/or bibliography addition is proposed to be be added to support the decreasing trend in the urinary concentrations of CPPDQ and 6PPDQ with increasing age.
5) Supplementary: Table S2. Please provide a footnote explaining the (%) as well as (±) symbols.
6) Supplementary: Table S2. The SRM ratios (average) can be added.
7) Supplementary: Table S4. Recovery range (%) can be added as well as equations and R2.
8) Please enrich current article with representative chromatograms.
Comments on the Quality of English Language
Minor editing of English language required.
Round 2
Reviewer 1 Report
Comments and Suggestions for Authors
Dear authors, I have noticed the modifications you have made. Although some drawbacks and limitations persist, from my point of view the editorial process can continue.
Author Response
Dear authors, I have noticed the modifications you have made. Although some drawbacks and limitations persist, from my point of view the editorial process can continue.
Response: we thank the reviewer very much for your comment.